# Therapeutic Effect of Calcimimetics on Osteoclast–Osteoblast Crosslink in Chronic Kidney Disease and Mineral Bone Disease

**DOI:** 10.3390/ijms21228712

**Published:** 2020-11-18

**Authors:** Kuo-Chin Hung, Jia-Feng Chang, Yung-Ho Hsu, Chih-Yu Hsieh, Mai-Szu Wu, Mei-Yi Wu, I-Jen Chiu, Ren-Si Syu, Ting-Ming Wang, Chang-Chin Wu, Lie-Yee Hung, Cai-Mei Zheng, Kuo-Cheng Lu

**Affiliations:** 1Division of Nephrology, Department of Medicine, Min-Sheng General Hospital, Taoyuan City 330, Taiwan; corey926@gmail.com (K.-C.H.); M001324@e-ms.com.tw (R.-S.S.); 2Division of Nephrology, Department of Internal Medicine, Taipei Medical University-Shuang Ho Hospital, New Taipei City 235, Taiwan; cjf6699@gmail.com (J.-F.C.); yhhsu@s.tmu.edu.tw (Y.-H.H.); maiszuwu@gmail.com (M.-S.W.); 09643@shh.org.tw (M.-Y.W.); 08888@s.tmu.edu.tw (I.-J.C.); masusuwin@yahoo.com.tw (L.-Y.H.); 3Division of Nephrology, Department of Internal Medicine, School of Medicine, College of Medicine, Taipei Medical University, Taipei 110, Taiwan; fish37435@Hotmail.com; 4Taipei Medical University-Research Center of Urology and Kidney (TMU-RCUK), School of Medicine, College of Medicine, Taipei Medical University, Taipei 110, Taiwan; 5Division of Nephrology, Department of Internal Medicine, Hsin Kuo Min Hospital, Taipei Medical University, Taoyuan City 320, Taiwan; 6Department of Nursing, Yuanpei University of Medical Technology, Hsinchu 300, Taiwan; 7Division of Nephrology, Department of Internal Medicine, En Chu Kong Hospital, New Taipei City 237, Taiwan; 8Renal Care Joint Foundation, New Taipei City 220, Taiwan; 9Department of Orthopaedic Surgery, School of Medicine, National Taiwan University, Taipei 106, Taiwan; dtorth76@yahoo.com.tw; 10Department of Orthopaedic Surgery, National Taiwan University Hospital, Taipei 106, Taiwan; 11Department of Orthopedics, En Chu Kong Hospital, New Taipei City 237, Taiwan; dtorth65@yahoo.com.tw; 12Department of Biomedical Engineering, Yuanpei University of Medical Technology, Hsinchu 300, Taiwan; 13Division of Nephrology, Department of Medicine, Taipei Tzu Chi Hospital, Buddhist Tzu Chi Medical Foundation, New Taipei City 231, Taiwan

**Keywords:** chronic kidney disease-mineral bone disease, cinacalcet, wnt, bone mineral density, osteoclast–osteoblast interaction, sclerostin, procollagen type 1 amino-terminal propeptide, tartrate-resistant acid phosphatase isoform 5b

## Abstract

We have previously demonstrated calcimimetics optimize the balance between osteoclastic bone resorption and osteoblastic mineralization through upregulating Wingless and int-1 (Wnt) signaling pathways in the mouse and cell model. Nonetheless, definitive human data are unavailable concerning therapeutic effects of Cinacalcet on chronic kidney disease and mineral bone disease (CKD-MBD) and osteoclast–osteoblast interaction. We aim to investigate whether Cinacalcet therapy improves bone mineral density (BMD) through optimizing osteocytic homeostasis in a human model. Hemodialysis patients with persistently high intact parathyroid hormone (iPTH) levels > 300 pg/mL for more than 3 months were included and received fixed dose Cinacalcet (25 mg/day, orally) for 6 months. Bone markers presenting osteoclast–osteoblast communication were evaluated at baseline, the 3rd and the 6th month. Eighty percent of study patients were responding to Cinacalcet treatment, capable of improving BMD, T score and Z score (16.4%, 20.7% and 11.1%, respectively). A significant correlation between BMD improvement and iPTH changes was noted (*r* = −0.26, *p* < 0.01). Nonetheless, baseline lower iPTH level was associated with better responsiveness to Cinacalcet therapy. Sclerostin, an inhibitor of canonical Wnt/β-catenin signaling, was decreased from 127.3 ± 102.3 pg/mL to 57.9 ± 33.6 pg/mL. Furthermore, Wnt-10b/Wnt 16 expressions were increased from 12.4 ± 24.2/166.6 ± 73.3 pg/mL to 33.8 ± 2.1/217.3 ± 62.6 pg/mL. Notably, procollagen type I amino-terminal propeptide (PINP), a marker of bone formation and osteoblastic activity, was increased from baseline 0.9 ± 0.4 pg/mL to 91.4 ± 42.3 pg/mL. In contrast, tartrate-resistant acid phosphatase isoform 5b (TRACP-5b), a marker of osteoclast activity, was decreased from baseline 16.5 ± 0.4 mIU/mL to 7.7 ± 2.2 mIU/mL. Moreover, C-reactive protein levels were suppressed from 2.5 ± 0.6 to 0.8 ± 0.5 mg/L, suggesting the systemic inflammatory burden may be benefited after optimizing the parathyroid–bone axis. In conclusion, beyond iPTH suppression, our human model suggests Cinacalcet intensifies BMD through inhibiting sclerostin expression and upregulating Wnt-10b/Wnt 16 signaling that activates osteoblastic bone formation and inhibits osteoclastic bone resorption and inflammation. From the perspective of translation to humans, this research trial brings a meaningful insight into the osteoblast–osteoclast homeostasis in Cinacalcet therapy for CKD-MBD.

## 1. Introduction

Chronic kidney disease (CKD) affects about 5–10% of the world population, and these patients are doomed to suffer from mineral and bone disorders (MBD) [1]. CKD-MBD represents a systemic disorder intricately involved in parathyroid–bone axis, including elevated circulating intact parathyroid hormone (iPTH), hypocalcemia, hyperphosphatemia, vitamin D deficiency, altered bone metabolism and ectopic soft tissue calcification [1]. Circulating iPTH levels underlie the homeostasis between bone formation and bone resorption, and secondary hyperparathyroidism (SHPT) due to mineral dysregulation occurs in parallel with CKD progression. In patients with CKD-MBD, SHPT plays a critical role in the pathogenesis of renal osteodystrophy (ROD), leading to abnormalities of bone turnover, mineralization, volume, linear growth, structure or strength [1]. Consequently, the risk of bone fracture is striking high in patients with hemodialysis (HD) and contributes to burden of mortality [2]. Cinacalcet is a calcimimetic agent that inhibits iPTH secretion by allosteric activation of the calcium-sensing receptor in the parathyroid gland. Oral daily administration of Cinacalcet is proven to reduce the risk of bone fracture and cardiovascular hospitalization in SHPT patients [3]. Our previous study in severe SHPT patients also demonstrated the triple combination therapy of Cinacalcet, cholecalciferol and calcitriol decreased the serum levels of iPTH and improved the bone mineral density (BMD) as well as 25-Hydroxyvitamin D (25(OH)D3) deficiency [4]. Our recent research reported Cinacalcet therapy was capable of strengthening both bone quality and quantity through stimulation of Wingless and int-1 (Wnt) signaling pathway in CKD mouse and in vitro [5]. If the above documented mechanisms are true, did Cinacalcet improve BMD through attenuating Wnt inhibitors to stimulate Wnt signaling, thereby inactivating osteoclasts and activating osteoblasts? Given that human data about Cinacalcet effect on osteocytic homeostasis have been unavailable, the last piece of the full picture in CKD-MBD therapy is still lacking.

The bone homeostasis is tightly regulated by the balance of the bioactivities of osteoblasts and osteoclasts. The bone resorption by osteoclasts exceeds the formation by osteoblasts in ROD, leading to bone loss and increased risk for osteoporotic events [6]. From the mechanistic view, Wnt signaling is a key pathway in osteocytes for bone formation that acts as an orchestrator of bone remodeling [7]. The Wnt ligands, such as Wnt-10b and Wnt 16, activate osteoblast differentiation and increase the rate of bone formation [5]. In contrast, sclerostin (SOST) as a potent Wnt inhibitor suppresses the differentiation of osteoblasts and attenuates bone formation [8]. Serum levels of SOST were significantly higher in patients with rheumatoid arthritis and correlated with disease severity, bone erosion and inflammatory markers such as C-reactive protein (CRP) [7]. Furthermore, procollagen type I propeptides (PINP) synthesized by active osteoblasts in the form of procollagen are the most collagen in mineralized bone [9]. Thus PINP is considered as a reliable marker of bone formation [10]. Compared with PINP, tartrate-resistant acid phosphatase isoform 5b (TRACP-5b) derived from osteoclasts circulates at elevated concentrations in various high bone turnover disorders with bone resorption [10,11]. By measuring the biomarkers of bone metabolism, the clinicians are able to monitor the risk of osteoporotic events in patients with CKD-MBD. Despite previously documented implications, definitive human data are unavailable concerning the intricate relationship between therapeutic effects of Cinacalcet on bone mineral density (BMD) and osteoclast–osteoblast interaction. In light of this, we aim to investigate whether Cinacalcet therapy improves BMD through optimizing osteoclast–osteoblast homeostasis in a human model. We hypothesized that the therapeutic effect of calcimimetic agent is mediated with downregulating the Wnt inhibitor SOST, activating downstream Wnt-10b and Wnt 16 signaling to promote osteoblastic bone formation (PINP) and reduce osteoclastic bone resorption (TRACP-5b), along with the improvement of inflammation marker CRP as an indicator of osteoarthritis.

## 2. Results

### 2.1. Bio-Demographic Characteristics of the Whole Study Population with Comparisons of Therapeutic Responses between Responders and Non-Responders after Six-Month Treatment of Cinacalcet

Forty patients with persistently high iPTH levels (>300 pg/mL for more than 3 months) were willing to participate in this open-label research trial. Baseline bio-clinical data of the whole study population with comparison between responders (n = 32) and non-responders (n = 8) were summarized in Table 1. The mean age was 54.8 ± 10.9 years (Male/Female: 27/13). Approximately 80% of HD patients were responding to fixed dose of Cinacalcet based on the results that BMD scores were increased after the treatment period of 6 months. HD vintage, baseline BMD, T score, Z score, baseline circulating levels of calcium, phosphate, albumin, hemoglobin, hematocrit, alkaline phosphatase, uric acid, triglyceride, total cholesterol, glycated hemoglobin (HbA1C), SOST, Wnt 10B, Wnt 16, PINP, TRACP-5b, CRP and 25(OH)D3 were not significantly different between two groups. Compared with the responder group, serum iPTH levels seemed to be higher in the non-responder group (694.1 ± 318.6 and 1012.2 ± 602.8 pg/mL, respectively). Figure 1 illustrated the Cinacalcet significantly improves parameters of BMD in the whole study population before and after six months of treatment, including BMD, BMD, T and Z scores (all *p*-values < 0.05). Table 2 demonstrated the comparison of therapeutic responses between responders and non-responders after the six-month treatment with Cinacalcet. Changes of BMD between responders and non-responders were 0.18 ± 0.33 and −0.04 ± 0.05, respectively (*p* < 0.01). Changes of T score between responders and non-responders were 0.27 ± 0.81 and −0.36 ± 0.46, respectively (*p* < 0.05). Changes of Z score between responders and non-responders were 0.21 ± 0.77 and −0.36 ± 0.40, respectively (*p* < 0.05). Figure 2 illustrated correlations between BMD and iPTH before (dash line) and after (solid line) six-month treatment of Cinacalcet. A significantly inverse correlation existed between BMD and iPTH before and after treatment (Pearson coefficient *r* = −0.23, *p* < 0.01; *r* = −0.26, *p* < 0.01; respectively). The slope of the linear regression was the average change in BMD for every one unit increase in iPTH. Compared with the pre-treatment slope, the post-treatment slope increases inversely, suggesting that BMD improvement is correlated with suppression of iPTH.

### 2.2. Baseline Lower iPTH Level Is Associated with Responsiveness to Cinacalcet Therapy

Table 3 demonstrated univariate logistic regression analysis of parameters associated with responsiveness to Cinacalcet treatment among HD patients with SHPT. At baseline, most parameters for CKD-MBD were not associated with Cinacalcet responsiveness. Nonetheless, baseline lower iPTH level (mean 694.1 ± 318.6 pg/mL) was associated with better responsiveness to Cinacalcet therapy.

### 2.3. Cinacalcet Treatment Attenuates the Expression of Wnt/β-Catenin Signaling Inhibitor SOST and Enhances the Activity of Wnt-10b/Wnt 16 at 3rd and 6th Months

As shown in Figure 3, SOST, an inhibitor of canonical Wnt/β-catenin signaling that reduced bone formation, was decreased from baseline 127.3 ± 102.3 pg/mL to 52.8 ± 33.3 and 57.9 ± 33.6 pg/mL at the 3rd and 6th month, respectively (*p* < 0.001). Furthermore, Wnt-10b/Wnt 16 expressions were increased from baseline 12.4 ± 24.2/166.6 ± 73.3 pg/mL to 33.7 ± 2.1/239.1 ± 60.4 pg/mL and 33.8 ± 2.1/217.3 ± 62.6 pg/mL at the 3rd and 6th month, respectively (all *p* values < 0.01).

### 2.4. Cinacalcet Treatment Upregulates PINP Expression to Activate Osteoblasts Responsible for Bone Formation and Downregulates TRACP-5b Expression to Inhibit Osteoclastic Bone Resorption Along with CRP Suppression at 3rd and 6th Months

As shown in Figure 4, PINP originated predominantly from proliferating osteoblasts as a marker of bone formation and osteoblastic activity was increased from baseline 0.89 ± 0.41 pg/mL to 91.78 ± 45.28 and 91.42 ± 42.34 pg/mL at the 3rd and 6th month, respectively (*p* < 0.001). In contrast, TRACP-5b derived from osteoclasts as the bone resorption marker and osteoclastic activity was decreased from baseline 16.5 ± 0.41 mIU/mL to 7.98 ± 2.58 and 7.72 ± 2.15 mIU/mL. Cinacalcet therapy suppressed C-reactive protein levels (from 2.5 ± 0.6 to 0.8 ± 0.5 mg/L), suggesting that inflammatory burden in CKD-MBD may be benefited after optimizing parathyroid–bone axis.

## 3. Discussion

Our research trial suggests that Cinacalcet intensifies BMD through attenuating the Wnt inhibitor SOST to stimulate Wnt 10b/Wnt16 signaling pathway, thereby inactivating osteoclastic bone resorption (TRACP-5b) and activating osteoblastic bone formation (PINP) along with the improvement of inflammatory marker CRP in addition to iPTH suppression (Figure 5). Several important issues deserve further discussion in this translational research trial.

In this study, 80% of HD patients with persistently high iPTH levels (mean value = 748.6 ± 389.6 pg/mL) for more than 3 months were responding to fixed dose of Cinacalcet based on the results that BMD, T and Z scores were improved during treatment period of 6 months. These outcomes were consistent with our previous study demonstrating the triple combination therapy of Cinacalcet, cholecalciferol and calcitriol decreased the serum iPTH level and improved the BMD among severe SHPT patients [4]. Nonetheless, changes in osteoblast–osteoclast biomarkers between pretreatment and posttreatment were not investigated. A similar but smaller scale of clinical study by Tsuruta et al. showed Cinacalcet increased the BMD of the femoral neck associated with the change of circulating levels of bone turnover marker alkaline phosphatase and bone-specific alkaline phosphatase in HD patients with SHPT [12]. Although the effect of calcimimetic therapy has been well-established, the intricate relationship among the Wnt inhibitor SOST, the Wnt signaling pathway, the osteoblastic bone formation marker PINP and the osteoclastic bone resorption marker TRACP-5b remains elusive in SHPT patients treated with Cinacalcet. Our studies provided direct evidence that Cinacalcet therapy fortified the BMD of HD patients with CKD-MBD, through optimizing the balance of osteoblastic and osteoclastic bioactivities. Cinacalcet therapy not only reduced the serum iPTH level, but also might benefit HD patients from the risk of cardiovascular events and bone fracture [13,14]. In light of this, Cinacalcet could serve as a comprehensive therapy for CKD-MBD.

For all baseline bio-clinical factors, only serum iPTH levels were higher in the non-responder group than the responder group (694.1 ± 318.6 and 1012.2 ± 602.8 pg/mL, respectively). To investigate this further, we used a univariate logistic regression analysis for parameters of CKD-MBD associated with responsiveness to Cinacalcet treatment (Figure 3). In Table 3, most CKD-MBD parameters could not predict the responsiveness to Cinacalcet. Notably, baseline lower iPTH level was associated with better Cinacalcet responsiveness, reflecting the treatment should be prompt and early in HD patients with SHPT. In contrast to previous studies, the Cinacalcet’s effects at different iPTH levels showed Cinacalcet was more effective in patients with more severe SHPT especially in those with iPTH ≥ 500 pg/mL [15]. Our data also showed that the post-treatment inverse correlation between BMD and iPTH was stronger than pre-treatment (Pearson coefficient *r* = −0.23, *p* < 0.01; *r* = −0.26, *p* < 0.01; respectively), providing the evidence that BMD improvement was correlated with suppression of iPTH.

Wnt signaling plays a critical regulatory role across bone modeling and bone remodeling, involving a lifelong process of osteogenesis and tissue renewal [16]. The Wnt signaling pathway consists of the canonical and the noncanonical pathways: the canonical pathway stabilizes β-catenin to mediate signaling and the noncanonical pathway acts independently of β-catenin [5]. Canonical Wnt ligands, particularly Wnt-10b and Wnt 16, are involved in bone homeostasis, including osteoblastic activity, bone formation and repression of osteoclastic activity and the role in cortical bone fractures [7,17,18,19,20]. In CKD-MBD, Wnt 16 and β-catenin genes are upregulated to activate inflammatory pathways along with oxidative stress, and bone formation signal like Wnt10b is downregulated [21]. Our prior research revealed that Cinacalcet increases osteoblastic mineralization, possibly through increased osteoclast Wnt-10b secretion [5]. In contrast, Wnt signaling antagonist SOST diminishes osteoblast differentiation and survival through binding low-density lipoprotein receptor-related protein 5 and 6 [16]. Antibodies against endogenous SOST have demonstrated promising results in promoting bone formation and fracture healing [22]. In accordance with the above findings in basic research, Wnt-10b and Wnt 16 levels were significantly elevated and SOST was biophysically decreased in our HD patients receiving Cinacalcet therapy.

Almost all CKD patients suffer from multiple morbidities, including cardiovascular diseases, which account for more than half of the deaths in HD patients [23,24]. From the perspective of SOST as an uremic vascular calcification inhibitor, previous study showed that higher serum level of SOST was intricately involved in lower short-term (18-month) cardiovascular mortality and inversely associated with serum iPTH and alkaline phosphatase levels in HD patients [25]. On the contrary, our data suggest better outcomes in BMD after Cinacalcet treatment by a decline in SOST levels and reciprocally up-regulating the Wnt pathway. SOST acts as an antagonist of Wnt signaling and thereby SOST suppression results in upregulation of Wnt-10b and Wnt16. In addition to down-regulation of SOST, our results addressed a simultaneously reduction of the indicator of osteoclast activity, TRACP-5b. Notably, there are two forms of TRACP circulating in human blood, TRACP-5a derived mainly by macrophages and dendritic cells, and TRACP-5b secreted mainly by osteoclasts. TRACP-5a was considered as a marker for chronic inflammation, whilst TRACP-5b was a marker for osteoclast number and the indicator for bone resorption [11,26]. Compared with healthy individuals, the serum TRACP-5a was significantly elevated and TRACP-5b was slightly increased in patients with rheumatoid arthritis, which manifested chronic inflammation and certain degree of bone destruction [27]. Similar results were noted in a randomized, open label study in chronic HD patients with SHPT receiving Cinacalcet therapy, whose serum levels of iPTH, FGF-23 and TRACP-5b were decreased after treatment [28]. Furthermore, the improvement of BMD was achieved through accentuating bone formation marker PINP and attenuating bone resorption marker TRACP-5b. A reduction in serum TRACP-5b in conjunction with an incremental change of serum PINP elucidated the osteoporotic fatalism in patients with CKD-MBD was reversed by Calcimimetic therapy along with the improvement of CRP as an indicator of bone inflammation. Our previous study disclosed that incremental serum levels of serum IL-6 in parallel with iPTH elevation could be effectively suppressed by calcitriol therapy, which indicated that hyperparathyroidism may aggravate inflammation in patients on maintenance HD [29]. Further study demonstrated calcitriol therapy could effectively reduce both inflammatory markers (CRP and IL-6) and oxidative stress through suppressing iPTH secretion, contributing to a decrement of inflammatory cytokines (CD4(+) IFN-γ) and an increment of anti-inflammatory cytokine (CD4(+) IL-4) [30]. In this study, Cinacalcet therapy suppressed CRP levels (from 2.5 ± 0.6 to 0.8 ± 0.5 mg/L), suggesting that systemic inflammatory burden in CKD-MBD may be benefited through optimizing parathyroid–bone axis. However, the detailed anti-inflammatory/antioxidant properties of Cinacalcet in alleviating osteoarthritis or systemic inflammation in CKD-MBD population require further investigation.

Qureshi et al. demonstrate that circulating SOST levels are associated with vascular calcification, implicating SOST serves as a surrogate marker between CKD-MBD and cardiovascular disease [31]. From then on, a large number of existing studies in the broader literature have examined the clinical use of SOST in CKD-MBD [32]. Nonetheless, the clinical use of SOST remains controversial. In many studies, higher serum SOST values are associated with fatal and nonfatal cardiovascular events, suggesting patients with a higher cardiovascular calcification scale have higher SOST serum levels [33,34]. On the contrary, elevated serum SOST levels were associated with better short-term cardiovascular outcomes in a large cohort study of 673 dialysis patients [25]. From these clues, we can speculate that the calcified vascular cells will secrete more SOST to process autocrine or paracrine manners to inhibit Wnt signaling effects on osteogenic trans-differentiation of vascular smooth muscle cells, preventing further calcification of the vessel wall [35]. Thus, circulating SOST levels should be interpreted according to clinical scenarios. In bone cases, osteocyte is the major source of SOST. The cross-sectional measurement of circulating SOST only reflects the dynamics of bone metabolism with particular reference to scenarios in which osteocytes may be perturbed [36]. Whether circulating the SOST level was bone-derived or vascular-derived in different bone turnover and vascular calcification status has yet to be elucidated. From the perspective of positive correlation between SOST and vascular calcification-related cardiovascular events, it would be reasonable to expect a decrease in SOST levels through attenuate the severity of vascular calcification after appropriate therapy, e.g., calcimimetics. Intriguingly, Kuczera P et al. reported cinacalcet treatment in hemodialysis patients with severe SHPT decrease serum PTH concentration and increases plasma SOST concentration, reflecting the more robust suppression of high bone turnover status in the study [37]. Compared with their study population receiving much higher therapeutic dose (maximal allowed dose of cinacalcet was 120 mg daily) for more advanced SHPT (mean iPTH level was 1138 pg/mL), our study subjects with a lower PTH level (748.6 pg/mL) were treated with a fixed low dose of oral Cinacalcet (25 mg/day), suggesting that different therapeutic strategies of cinacalcet result in different changes of circulating SOST.

We recognize several limitations of our study. To begin with, the sample size in our human model was relatively small and all our patients were Asian people, limiting the power of generalization to other populations as standardized treatment suggestion. Next, all study subjects were treated with a fixed low dose of oral Cinacalcet (25 mg/day) without dose uptitration in the present study, limiting the maximum effect of Cinacalcet therapy. Meanwhile, the efficacy of long-term Cinacalcet use, severe adverse effects or toxicity still requires a multi-factored clinical trial approach and longer follow-up period to monitor. Finally, the dual X-ray absorptiometry (DXA) scan using lumbar spine BMD was interfered with a highly prevalent abdominal aorta calcification in our HD patients, leading to higher values of BMD, T and Z score. Therefore, these results should be interpreted with caution.

In conclusion, beyond iPTH suppression, our human model suggests Cinacalcet intensifies BMD through inhibiting SOST expression and upregulating Wnt-10b/Wnt 16 signaling that activate osteoblastic bone formation (PINP) and inactivate osteoclastic bone resorption (TRACP-5b), along with the improvement of bone inflammation after optimizing parathyroid–bone axis. Further corresponding investigation focusing on post-treatment changes of human bone histomorphometry in CKD-MBD patients is warranted. Furthermore, our data provide molecular therapeutic targets for CKD-MBD that Wnt agonists, SOST antagonists and novel bone biomarkers as surrogate markers merit future research. From the perspective of translation to humans, this research trial brings a meaningful insight into the osteoblast–osteoclast homeostasis in Cinacalcet therapy for CKD-MBD.

## 4. Materials and Methods

### 4.1. Study Design and Patient Eligibility

The research trial was designed as an open-label study. The study had been approved by the Research Ethics Review Committee of Taipei Medical University (TMU-IRB-N201705021) and Min-Sheng General Hospital (MSGH-IRB-N2018004) in accordance with the ethical standards of the committee and the Helsinki declaration for research in humans, also the clinical trial ID was registrated. The written informed consent was obtained from all the participants before the study. Patients older than 18 years of age were eligible for enrollment. All enrolled patients had to receive thrice-weekly MHD for at least 3 months. Based on our study interest, SHPT patients with persistently high serum iPTH > 300 pg/mL even after more than 3 months of calcitriol treatment were included. Then, all included patients needed to stop active vitamin D therapies at least 30 days prior to entering the study. In contrast, SHPT patients were excluded if they were using calcimimetic agents, allergic to calcimimetics, pregnant, breastfeeding or of childbearing potential and not practicing birth control. SHPT patients were also excluded from the study because of terminal illness, bed-ridden status, active infections, advanced cancer, active hepatitis, severe protein-energy wasting, incomplete data, unwilling to participate or major cardiovascular events within 3 months of the enrollment.

### 4.2. Treatment Intervention

All study subjects were treated with a fixed dose of oral Cinacalcet (CINACA^®^, Anxo, Taiwan) (25 mg/day) at baseline in the present study. The doses of calcium-based and other phosphate binders could be adjusted according to clinical laboratory values throughout the study. Calcium-based phosphate binders could be increased when the serum calcium was less than 8.4 mg/dL or study subjects suffer from symptomatic hypocalcemia. Cinacalcet therapy was given during the whole study period and was withdrawn if the serum iPTH was persistently lower than 150 pg/mL for 4 weeks or symptomatic hypocalcemia < 8.4 mg/dL despite the supplement of calcium-containing phosphate binders and/or vitamin D sterols. Adverse events were collected from patients’ reports and in response to non-directed questioning at each study visit.

At the beginning of study, 52 secondary hyperparathyroid patients with persistently high serum iPTH > 300 pg/mL even after more than 3 months of calcitriol treatment were included. In total, 12 patients were dropped out during the study (loss of follow up (N: 3), declined for dose adjustment (N: 2), hospitalization with infection condition (N: 3) and gastrointestinal, GI, intolerance (N: 4)). In our study population, 9 of our patients suffered from GI side effects and included in dropped out cases, reflux esophagitis (N: 3), abdomen pain (N: 4) and diarrhea (N: 2). No significant hypocalcemia is experienced in most of our study patients.

### 4.3. Patient Demographic Information and BMD Measurement

All study subjects initially used low calcium (2.5 mEq/L) dialysate for HD with adjustments according to patient tolerance and serum calcium levels throughout the study. A baseline visit was performed just before the start of the study, and further study visits were performed at 4, 8, 12, 16, 20 and 24 weeks. HD vintage was defined as the duration of time between the first day of HD treatment and the first day that the patient entered the cohort. The BMD, T and Z score of lumbar spine (LS) were determined by DXA scan before and at the end of the study.

### 4.4. Serum Biochemical and Bone Metabolism Parameters

Pre-dialysis blood samples were obtained from the existing vascular access for further analysis. Following standard centrifugation within 1 h of collection, serum was aliquoted and immediately frozen at −80 °C for storage until analysis. Serum biochemical parameters and bone turnover markers were assessed according to the manufacturer’s instructions at baseline and during follow-up. CRP, calcium, phosphate, alkaline phosphatase, albumin, uric acid, triglyceride, total cholesterol, HbA1C, hematocrit and hemoglobin were measured using standard laboratory techniques. Serum iPTH levels were measured in an immunoradiometric assay (Nichols Institute Diagnostics, San Juan Capistrano, CA, USA). Serum 25(OH)D3 was determined by enzyme-linked immunosorbent assay (ELISA) kit (Immundiagnostik AG, Bensheim, Germany). Serum SOST, Wnt-10b and Wnt-16 was measured using ELISA kits (Cusabio Technology LLC., Houston, TX, USA). Serum PINP, a marker of bone formation was determined by ELISA kit (USCN Life Science Inc., Wuhan, China). Serum TRACP-5b, a marker of bone resorption, was measured by ELISA kit (Quidel, San Diego, CA, USA).

### 4.5. Objectives and Outcome Measuresemnt

Our objective aims to investigate whether Cinacalcet therapy improves BMD through optimizing osteoclast–osteoblast homeostasis in this human model. Beyond the effect of lowering iPTH, we hypothesized that the therapeutic effect of the Calcimimetic agent was mediated with downregulating the Wnt inhibitor SOST, activating downstream Wnt-10b and Wnt 16 signaling to promote osteoblastic bone formation (PINP) and reduce osteoclastic bone resorption (TRACP-5b), along with the improvement of general inflammation marker CRP. We assessed therapeutic effects of Cinacalcet on BMD changes as the outcome measurement. The responders were defined as the improvement in BMD of LS (post-treatment BMD–pre-treatment BMD = BMD change > 0) determined by DXA scan.

### 4.6. Statistical Analysis of Data

All data are expressed as the mean ± SD using the GraphPad Prism Program (GraphPad, San Diego, CA, USA) or PASW Statistics SPSS version 22.0 (IBM, NY, USA). The linear regression and multivariate analysis were performed by SPSS version 22.0 (IBM, NY, USA). Quantitative data were analyzed with a non-paired or paired Student’s *t*-test. The significance threshold was set at 5% (*p <* 0.05). The G power software and *t*-test were used to estimate the number of samples in this study.

## Figures and Tables

**Figure 1 ijms-21-08712-f001:**
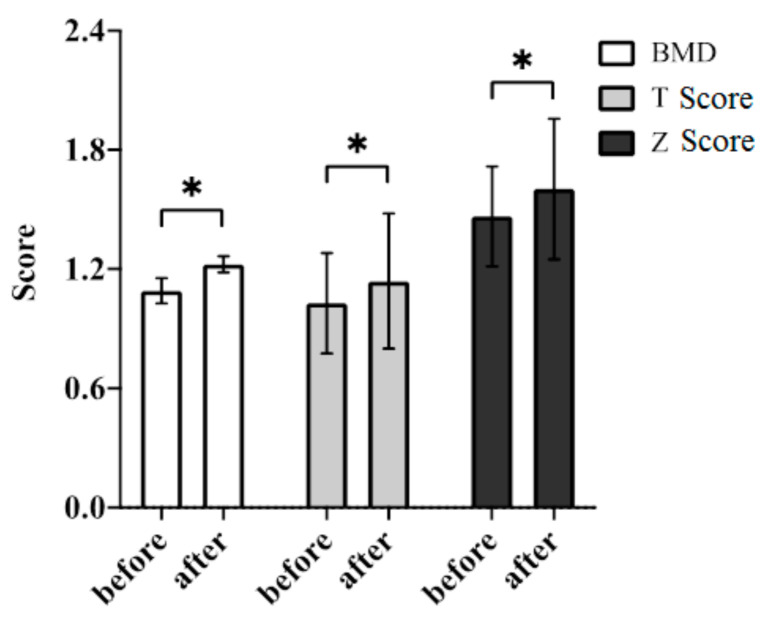
Therapeutic effects of Cinacalcet on parameters of BMD in the whole study population. Note: * represents the *p*-value < 0.05 to compare the differences between the two indicated groups by Mann–Whitney Rank Sum Test.

**Figure 2 ijms-21-08712-f002:**
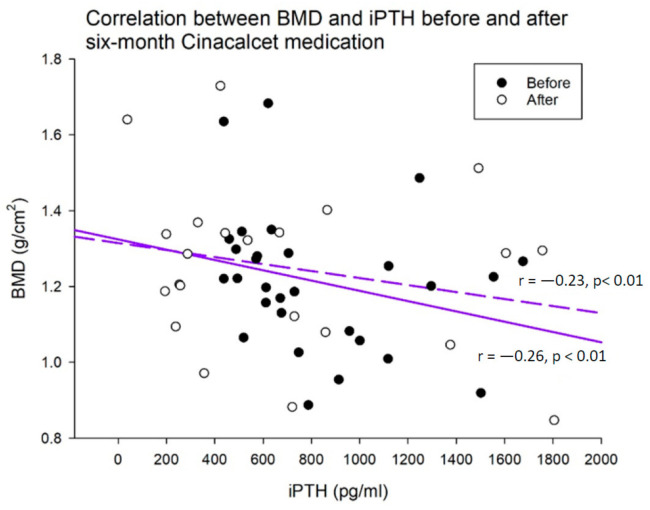
Correlations between BMD and iPTH before (dash line) and after (solid line) six-month treatment of Cinacalcet. A significantly inverse correlation exists between BMD and iPTH before and after treatment. Compared with the pre-treatment slope, the post-treatment slope increases inversely, suggesting that BMD improvement is correlated with suppression of iPTH.

**Figure 3 ijms-21-08712-f003:**
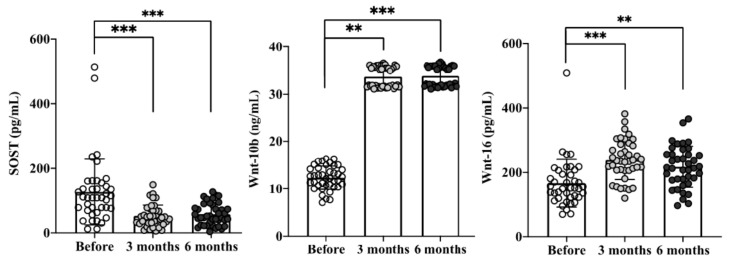
Cinacalcet treatment attenuates the expression of Wnt/β-catenin signaling inhibitor SOST and enhances the activity of Wnt-10b/Wnt 16 at 3rd and 6th months. Quantitative data were analyzed with a paired Student’s *t*-test. ** *p* < 0.01; and ****** p* < 0.001 to compare the differences between the two indicated groups.

**Figure 4 ijms-21-08712-f004:**
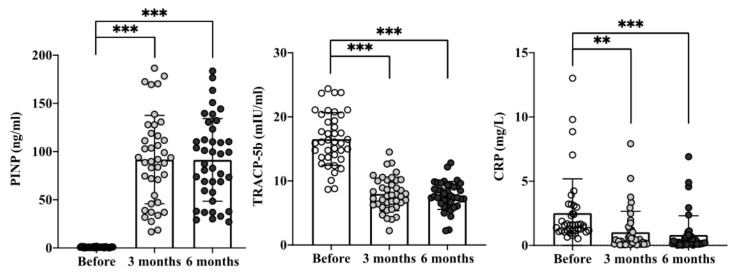
Cinacalcet treatment upregulates PINP expression to activate osteoblasts responsible for bone formation and downregulates TRACP-5b expression to inhibit osteoclastic bone resorption along with CRP suppression at the 3rd and 6th month. Quantitative data were analyzed with a paired Student’s *t*-test. ** *p* < 0.01; and ****** p* < 0.001 to compare the differences between the two indicated groups.

**Figure 5 ijms-21-08712-f005:**
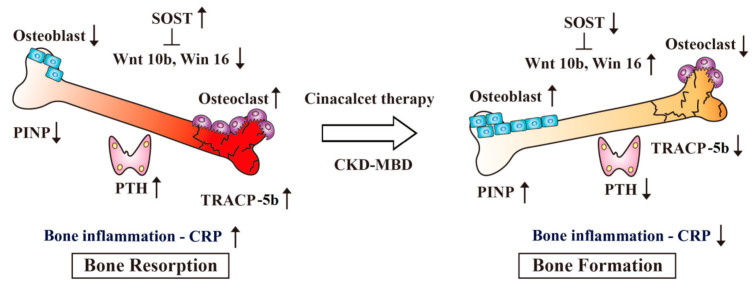
Potential mechanisms of therapeutic effects of Cinacalcet on CKD-MBD. The schematic diagram illustrates that increased SOST expression in patients with secondary hyperparathyroidism inhibits Wnt 10b/Wnt 16 signaling pathway, leading to activating osteoclastic bone resorption (TRACP-5b) and inactivating osteoblastic bone formation (PINP), and ultimately bone inflammation and low bone density. Beyond iPTH suppression, Cinacalcet therapy intensifies bone density through inhibiting SOST expression and upregulating Wnt-10b/Wnt 16 signaling that activates osteoblastic bone formation (PINP) and inactivates osteoclastic bone resorption (TRACP-5b), along with the improvement of inflammatory milieu (CRP). CKD-MBD, chronic kidney disease and mineral bone disease; CRP, C-reactive protein; iPTH, intact parathyroid hormone; PINP, procollagen type I propeptides; SOST, sclerostin; TRACP-5b, tartrate-resistant acid phosphatase isoform 5b; Wnt, Wingless and int-1; ↑, increase; ↓, decrease.

**Table 1 ijms-21-08712-t001:** Baseline bio-demographic characteristics of the whole study population with comparison between responders and non-responders.

Variables	Overall (n = 40)	Responders (n = 32)	Non-Responders (n = 8)	*p*-Value
	mean ± SD	mean ± SD	mean ± SD	
Age (years)	54.8 ± 10.9	55.4 ± 11.3	50.8 ± 7.7	0.25
Male, n (%)	27 (67.5%)	22 (64.7%)	5 (83.3%)	0.35
HD duration (years)	6.5 ± 5.6	6.3 ± 5.9	7.3 ± 4.1	0.68
Baseline BMD (g/cm^2^)	1.1 ± 0.4	1.1 ± 0.4	1.1 ± 0.2	0.93
T score	1.0 ± 1.5	1.3 ± 1.5	1. ± 0.2	0.93
Z score	1.1 ± 0.4	1.1 ± 0.4	1.1 ± 0.2	0.93
iPTH (pg/mL)	748.6 ± 389.6	694.1 ± 318.6	1012.2 ± 602.8	0.03
Calcium (mg/dL)	9.1 ± 0.8	9.1 ± 0.8	9.0 ± 0.8	0.60
Phosphate (mg/dL)	5.6 ± 1.4	5.7 ± 1.4	5.6 ± 1.8	0.89
Albumin (mg/dL)	3.8 ± 0.4	3.8 ± 0.4	3.8 ± 0.2	0.85
Hematocrit (%)	32.0 ± 3.2	32.0 ± 3.4	32.0 ± 2.4	0.73
Hemoglobin (g/dL)	10.8 ± 1.2	10.7 ± 1.3	10.9 ± 0.6	0.99
ALK-P (IU/L)	111.3 ± 87.3	109.9 ± 89.0	119.3 ± 85.8	0.81
Uric acid (mg/dL)	7.3 ± 1.5	7.2 ± 1.5	7.5 ± 1.4	0.71
Triglyceride (mg/dL)	176.8 ± 126.1	185.8 ± 136.0	116.3 ± 38.8	0.23
T-Cholesterol (mg/dL)	157.8 ± 43.6	158.8 ± 46.9	153.0 ± 23.7	0.66
HbA1C (%)	7.1 ± 1.9	7.6 ± 2.1	6.2 ± 1.3	0.18
SOST (pg/mL)	127.3 ± 103.7	133.3 ± 107.3	94.3 ± 80.3	0.33
Wnt 10B (pg/mL)	123.5 ± 24.5	122.5 ± 25.1	120.9 ± 22.4	0.54
Wnt 16 (pg/mL)	166.5 ± 74.3	168.2 ± 78.0	157.3 ± 52.8	0.68
PINP (ng/mL)	0.9 ± 0.4	0.9 ± 0.4	0.8 ± 0.4	0.59
TRACP-5b (IU/L)	16.5 ± 0.4	16.6 ± 0.4	16.4 ± 0.4	0.90
CRP (mg/L)	2.5 ± 2.7	2.6 ± 2.8	2.1 ± 1.1	0.46
25(OH)D3 (ng/mL)	13.2 ± 9.6	12.7 ± 9.7	15.7 ± 9.2	0.49

Data are expressed as the mean ± SD or n (%). Boldface indicates where the values differ significantly between responders and non- responders. Quantitative data were analyzed with a non-paired Student’s *t*-test. 25(OH)D3, 25-Hydroxyvitamin D; ALK-P, alkaline phosphatase; BMD, bone mineral density; CRP, C-reactive protein; HD, hemodialysis; HbA1C, glycated hemoglobin; iPTH, parathyroid hormone; PINP, procollagen type I propeptides; SD, standard deviation; SOST, sclerostin; T-Cholesterol, total cholesterol; TRACP-5b; tartrate-resistant acid phosphatase isoform-5b.

**Table 2 ijms-21-08712-t002:** Comparison of therapeutic responses between responders and non-responders after six-month treatment of Cinacalcet.

	Responders (n = 32)	Non-Responders (n = 8)	*p*-Values
BMD changes	0.18 ± 0.33	−0.04 ± 0.05	<0.05
T score changes	0.27 ± 0.81	−0.36 ± 0.46	<0.05
Z score changes	0.21 ± 0.77	−0.36 ± 0.40	<0.05

Data are expressed as mean ± standard deviation. Quantitative data were analyzed with a non-paired Student’s *t*-test.

**Table 3 ijms-21-08712-t003:** Univariate logistic regression analysis of parameters associated with responsiveness to Cinacalcet treatment among hemodialysis (HD) patients with secondary hyperparathyroidism (SHPT).

Variable	Univariate OR for Responsiveness (95% CI)	*p*-Value
iPTH (pg/mL)	0.98 (0.96–1.00)	0.05
ALK-P (IU/L)	1.00 (0.99–1.01)	0.80
Calclium (mg/dL)	1.10 (0.34–3.55)	0.87
Phosphate (mg/dL)	1.05 (0.56–1.96)	0.88
SOST (pg/mL)	1.01 (0.99–1.02)	0.40
Wnt 10B (pg/mL)	0.99 (0.95–1.03)	0.55
Wnt 16 (pg/mL)	1.00 (0.99–1.02)	0.74
PINP (ng/mL)	2.19 (0.18-6.71)	0.54
TRACP-5b (IU/L)	1.15 (0.14–9.86)	0.90
CRP (mg/L)	1.09 (0.72–1.66)	0.68
25(OH)D3 (ng/mL)	0.97 (0.89–1.06)	0.47

Boldface indicates where the values differ significantly between responders and non-responders (*p* = 0.046). 25(OH)D3, 25-Hydroxyvitamin D; ALK-P, alkaline phosphatase; CRP, C-reactive protein; iPTH, parathyroid hormone; PINP, procollagen type I propeptides; SOST, sclerostin; TRACP-5b; tartrate-resistant acid phosphatase isoform-5b.

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
