# Peer review of "Therapeutic Effect of Calcimimetics on Osteoclast–Osteoblast Crosslink in Chronic Kidney Disease and Mineral Bone Disease"

_ijms, 2020, doi:10.3390/ijms21228712_

Round 1

Reviewer 1 Report

The authors's results suggest Cinacalcet intensifies BMD through inhibiting sclerostin expression and upregulating Wnt-10b/ Wnt 16 signaling that activates osteoblastic bone formation and inhibits osteoclastic bone resorption and inflammation in human model. This study is well designed and well performed. The data sufficiently support the conclusion. I only have minor concerns.

  1. In Fig2. the legent "after" should be "After".
  2.  Figure 6 or Figure 5, please double check.
  3. Carefully check your statistic and make all the tests are right. Especially in Figure 1, it is difficult to convince the significan difference.
  4.  The authors mentioned that they had finish the animal study and cell study. Are they animal cells or human cells? If they are animal cells, you should redo the experiment using human cells to make sure the mechanism study is right.

Author Response

Response to Reviewer 1 Comments

The authors's results suggest Cinacalcet intensifies BMD through inhibiting sclerostin expression and upregulating Wnt-10b/ Wnt 16 signaling that activates osteoblastic bone formation and inhibits osteoclastic bone resorption and inflammation in human model. This study is well designed and well performed. The data sufficiently support the conclusion. I only have minor concerns.

  1. In Fig2. the legend "after" should be "After".

Response 1: We changed it and remade a new figure.

  1. Figure 6 or Figure 5, please double check.

Response 2: Great thanks for the correction. We did appropriate correction to Figure 5.

  1. Carefully check your statistic and make all the tests are right. Especially in Figure 1, it is difficult to convince the significant difference.

Response 3: The represents the P-value < 0.05 to compare the differences between the two indicated groups by Mann-Whitney Rank Sum Test. We will correct figure 1 description.

  1. The authors mentioned that they had finish the animal study and cell study. Are they animal cells or human cells? If they are animal cells, you should redo the experiment using human cells to make sure the mechanism study is right.

Response 4: We used primary cell cultures of osteoclasts derived from stimulation of 8-week-old male C57BL/6 mice bone marrow-derived monocytes (osteoclast precursors) with 50 ng/mL macrophage colony stimulating factor (M-CSF) and receptor activation of NF-κB ligand (RANKL) in our previous study (Zheng et al. 2019*). We did not use human osteoclast precursors derived from peripheral blood monocytes as a source for osteoclasts. Similarly, we could not obtain human bone marrow-derived monocytes as a source for osteoclasts. Indications for bone marrow aspiration in our hospital as follows: 1. Unexplained anemia, leukopenia, thrombocytopenia, or pancytopenia. 2. Lymphoma or solid tumors. 3. Elevated peripheral counts (eg, polycythemia, thrombocytosis, leukocytosis). 3. Plasma cell disorders and leukemias. 4. Iron metabolism and stores. 5. Deposition and storage diseases (eg, amyloidosis, Gaucher disease). 6. Fever of undetermined origin. 7. Unexplained splenomegaly 8. Chromosomal disorders in neonates. 9. A potential allogeneic hematopoietic cell donor. Therefore, our institutional review board (IRB) committee forbids such primary cell cultures of osteoclasts and osteocytes derived from human bone marrow due to ethical issues in clinical research. Currently, commercial human osteoclasts and osteoblasts have not been available for purchase. In light of your suggestions, we plan to persuade IRB committee to allow us to use human osteoclast precursors in future studies.

* Zheng CM, Hsu YH, Wu CC et al. (2019) Osteoclast-Released Wnt-10b Underlies Cinacalcet Related Bone Improvement in Chronic Kidney Disease. Int J Mol Sci 20 doi:10.3390/ijms20112800

Reviewer 2 Report

Review of “Therapeutic Effect of Calcimimetics on Osteoclast-Osteoblast Crosslink in Chronic Kidney Disease and Mineral Bone Disease“

This study investigated the effect of Calcimimetics on osteoporosis in patients with HD. This study revealed that Calcimimetics had positive effect on osteoporosis and that the positive effects were more pronounced in patients with low iPTH. This study is interesting topic, I have some questions.

  1. The authors said that “the research trial was designed as an open-label study.” This reviewer thinks that because this study is a clinical trial, registration must be needed. Please show registration number.
  2. How to calculate the sample size?
  3. None of the participants drop out?
  4. How about the side effect?

Author Response

Response to Reviewer 2 Comments

This study investigated the effect of Calcimimetics on osteoporosis in patients with HD. This study revealed that Calcimimetics had positive effect on osteoporosis and that the positive effects were more pronounced in patients with low iPTH. This study is interesting topic; I have some questions.

  1. The authors said that “the research trial was designed as an open-label study.” This reviewer thinks that because this study is a clinical trial, registration must be needed. Please show registration number.

Response 1: We did not register in clinical trial.com. However, our Institutional. Review Board, IRB number in Taipei Medical University and Min-Sheng General Hospital are N201705021 and N2018004. We add this in our revised manuscript as follows:                                                                            The study had been approved by the Research Ethics Review Committee of Taipei Medical University (TMU-IRB-N201705021) and Min-Sheng General Hospital (MSGH-IRB-N2018004) in accordance with the ethical standards of the committee and the Helsinki declaration for research in humans. Line 336

  1. How to calculate the sample size?

Response 2: The G power software and t-test were used to estimate the number of sample in this study. Under the setting of alpha = 0.05, power = 80%, effect size = 0.4, and two-tailed test, the minimum sample size of 52 was required. We add this in our revised manuscript. Line 406

  1. None of the participants drop out?

Response 3: Based on our study interest, 52 secondary hyperparathyroid patients with persistently high serum iPTH > 300 pg/mL even after more than 3 months of calcitriol treatment were included. 12 patients were dropped out during the study (loss of follow up(N:3), declined for dose adjustment(N:2), hospitalization with infection condition(N:3) and GI intolerance(N:4)). We add this in our revised manuscript. Line 363

  1. How about the side effects?

Response 4: 9 of our patients suffered from GI side effects and included in dropped out cases., reflux esophagitis (N: 3), abdomen pain (N: 4) and diarrhea (N: 2). No significant hypocalcemia is experienced in most of our study patients. We add this in our revised manuscript. Line 366

Round 2

Reviewer 1 Report

I have no more concerns.

Author Response

Great thanks for reviewer.

Reviewer 2 Report

Ethical concerns remain. I recognize that this is an interventional study. I am aware that any interventional study must be registered in a clinical research database. Please point out if my recognition about registration of interventional study is wrong.

Author Response

Thanks for reviewer,

We already register in clinical trial.com. We will add NCT ID number in further our revised manuscript as follows: Line 341

Round 3

Reviewer 2 Report

No further comments.